# Probing optical anapoles with fast electron beams

Carlos Maciel-Escudero[1,2,8], Andrew B. Yankovich [3,8], Battulga Munkhbat [3,4], Denis G. Baranov [3,5], Rainer Hillenbrand [2,6], Eva Olsson [3] ✉, Javier Aizpurua [1,7] ✉ & Timur O. Shegai [3] ✉

Optical anapoles are intriguing charge-current distributions characterized by a strong suppression of electromagnetic radiation. They originate from the destructive interference of the radiation produced by electric and toroidal multipoles. Although anapoles in dielectric structures have been probed and mapped with a combination of near- and far-field optical techniques, their excitation using fast electron beams has not been explored so far. Here, we theoretically and experimentally analyze the excitation of optical anapoles in tungsten disulfide ($WS_2$) nanodisks using Electron Energy Loss Spectroscopy (EELS) in Scanning Transmission Electron Microscopy (STEM). We observe prominent dips in the electron energy loss spectra and associate them with the excitation of optical anapoles and anapole-exciton hybrids. We are able to map the anapoles excited in the $WS_2$ nanodisks with subnanometer resolution and find that their excitation can be controlled by placing the electron beam at different positions on the nanodisk. Considering current research on the anapole phenomenon, we envision EELS in STEM to become a useful tool for accessing optical anapoles appearing in a variety of dielectric nanoresonators.

Anapoles are particular charge-current distributions giving rise to an optical phenomenon characterized by strong suppression of electromagnetic radiation[1,2]. This phenomenon is typically understood as the interference between the electromagnetic (EM) field produced by a Cartesian electric multipole and the EM field of a toroidal multipole. When this interference is destructive, the system yields a non-radiating current configuration known as the optical anapole state[3,4]. Excitation of the anapole in a polarizable nanostructure greatly suppresses its scattering cross section thus providing invisibility to nanoobjects[5], which offers promising applications in nanophotonics[6–10]. Additionally, optical anapoles concentrate EM fields inside the nanoresonators serving to enhance nonlinear harmonic generation[11,12], four-wave mixing[13], Raman scattering[14,15], and photothermal nonlinearities[16].

Intense experimental and theoretical efforts have been devoted to identifying optical anapole states in different dielectric nanostructures. However, the detection of ideal optical anapoles is complicated and usually requires the suppression of other multipole resonances. Typically, this suppression is achieved by engineering the nanoresonator geometry and by structuring the incident illumination[6,7,10]. Complementary to far-field characterization, understanding how anapoles are excited by localized probes can be of paramount importance in order to control and realize the full potential of these non-radiating states. In this context, Scanning Near-Field Optical Microscopy (SNOM) was applied to study the near-field distribution of Si disks at relevant wavelengths[5,9]. In the experiment discussed in ref. 5, for example, the field around the disk was mapped at

[1]Materials Physics Center, CSIC-UPV/EHU, Paseo de Manuel Lardizabal, Donostia-San Sebastián 20018, Spain. [2]CIC NanoGUNE BRTA and Department of Electricity and Electronics, Tolosa Hiribidea, Donostia-San Sebastián 20018, Spain. [3]Department of Physics, Chalmers University of Technology, 41296 Göteborg, Sweden. [4]Department of Photonics Engineering, Technical University of Denmark, Kgs. Lyngby, Copenhagen 2800, Denmark. [5]Center for Photonics and 2D Materials, Moscow Institute of Physics and Technology, Dolgoprudny 141700, Russia. [6]IKERBASQUE, Basque Foundation for Science, Bilbao 48011, Spain. [7]Donostia International Physics Center, Paseo de Manuel Lardizabal, Donostia-San Sebastián 20018, Spain. [8]These authors contributed equally: Carlos Maciel-Escudero, Andrew B. Yankovich. ✉e-mail: eva.olsson@chalmers.se; aizpurua@ehu.eus; timurs@chalmers.se

the anapole wavelength, revealing a maximum in the amplitude of the near field in the middle of the disk. In the second experiment, discussed in ref. 9, it was found that the normal component of the electric field induced at the disks is reduced at the anapole wavelength. SNOM offers the advantage of spatially mapping both the amplitude and phase of near-field patterns, however, the spatial resolution is limited by the dimensions of the near-field probes (an optical fiber or a metallic tip), which are typically of the order of tens of nanometers. More importantly, the near-field probe itself can couple to the sample[17], leading to spectral shifts of the resonance modes and potential distortions when probing the anapoles. In contrast to SNOM techniques, Electron Energy Loss Spectroscopy (EELS) in Scanning Transmission Electron Microscopy (STEM) is a non-disturbing technique that accesses the clean electromagnetic fields and modes of a sample. At the same time, EELS allows for accessing not only electric dipole modes but also quadrupoles and higher-order modes that do not, or weakly, couple to far-field radiation.

EELS in STEM employs a tightly focused electron beam that allows for mapping the optical properties of a material with tens of meV energy resolution and down to atomic scale spatial resolution while simultaneously relating this information to the sample's precise size, shape, and structure[18,19]. For example, EELS has been used to characterize and map localized plasmons in different nanostructures[20–23], to probe electronic excitations such as excitons in Transition Metal Dichalcogenide (TMDC) materials[24], to predict optical toroidal modes in silver heptamer cavities[25], and to map photonic modes of dielectric silicon nanocavities[26]. Recently, EELS has also proven useful to resolve strong light-matter interactions with unprecedented spatial and spectral resolution[27–31]. However, a detailed study of optical anapoles excited by fast electrons is still pending.

Here, we present a numerical and experimental analysis of optical anapoles and anapole-exciton hybrids in $WS_2$ nanodisks using EELS. We first identify optical anapole states in model high-index dielectric nanodisks by calculating the electron energy losses experienced by a focused electron beam traveling in an aloof trajectory in the vicinity of the nanodisk. We demonstrate that the destructive interference between electric and toroidal Cartesian multipoles induced in the nanodisk manifests as dips in the electron energy loss (EEL) spectra. Optical anapoles are typically found in high-index dielectric nanostructures. To experimentally access the spectral and spatial information about anapoles, we fabricate nanodisks from exfoliated multilayer TMDC $WS_2$ with a high refractive index in the visible and infrared ranges[32–34], even larger than that of standard Si and other semiconductors which are usually employed to fabricate anapole resonators. To experimentally reveal and map anapole excitations in EELS, we then perform monochromated EELS experiments on the $WS_2$ nanodisks and compare experimental EEL spectra with numerical calculations. We find that by varying the nanodisk dimensions, the anapoles supported by the $WS_2$ nanodisk can be tuned to match the resonance of an excitonic transition of $WS_2$, leading to anapole-exciton hybridization. The emergence of this light-matter interaction regime shows another advantage of using TMDCs instead of other conventional high-index materials. Finally, we show the possibility to spatially map optical anapoles and to control their excitation by placing the electron beam at different positions on the nanodisk, demonstrating the potential of EELS to access these special non-radiating charge-current distributions.

## Results

### Theoretical prediction of optical anapoles in EEL spectra

We begin our study by describing the general features appearing in the EEL spectra of a model high-index dielectric nanodisk that can exhibit optical anapole states[35]. Figure 1a illustrates a sketch of the system under study: a dielectric disk excited by an electron beam traveling in aloof trajectory close to the disk. We choose a disk of variable radius $R$,

thickness $d = 55$ nm and permittivity $\varepsilon = 18$. The electron beam travels with velocity $v = 0.7c$ (200 keV, $c$ being the speed of light in vacuum) along the $z$-axis at a distance $b = 1.1R$ with respect to the nanodisk center (impact parameter). We calculate the EEL probability $\Gamma(\omega)$ numerically using the classical dielectric theory[18,36] (see "Methods" and Supplementary Note 1).

Figure 1b shows the calculated $\Gamma(\omega)$ spectra for different nanodisk radius $R$ in the energy range of 0.5 eV to 2.5 eV. The spectra feature a series of peaks (white dotted lines) and dips (gray dashed lines) that monotonously shift to higher energies as the nanodisk radius $R$ reduces from 300 nm to 100 nm. The positions of these peaks and dips do not vary with the electron beam velocity (see Supplementary Note 2). These peaks can be associated with the different resonant modes of the nanodisk excited by the electron beam (see Supplementary Note 3). In the following, we will focus on the underlying physics of the spectral dips and their relationship with the anapole phenomenon. We label these dips as $A_{ij}^E$, with $i$ and $j$ being integer index labels, based on symmetry reasons that will be explained below.

The anapole phenomenon originates from the destructive interference between the electric and toroidal Cartesian multipole moments inside the nanodisk with identical amplitude and far-field patterns. Due to this destructive interference, the radiation emitted by the disk is suppressed, and thus the anapole phenomenon manifests as a dip in the scattering cross section of the disk. To corroborate that the dips in the EEL spectra are associated with anapole excitation by fast electron beams, we analyze the EEL probability of the 250 nm radius nanodisk (blue line in Fig. 1c, where the red dots indicate the spectral dips). To that end, we perform a multipole decomposition of the current density $\mathbf{J}_{ind}(\mathbf{r})$ induced by the electron beam in the nanodisk. The induced current density $\mathbf{J}_{ind}(\mathbf{r})$ can be described by a series of exact (spherical) multipole moments induced in the disk. In the long-wavelength approximation each multipole moment can be expressed as a superposition of the so-called Cartesian multipole moments[37–39]. Particularly, the spherical electric dipole moment $\mathbf{P}_{sph}(\omega)$ is approximated by the following superposition of the electric, $\mathbf{P}_{car}(\omega)$, and toroidal, $\mathbf{T}_{car}(\omega)$, Cartesian dipole moments: $\mathbf{P}_{sph}(\omega) \approx \mathbf{P}_{car}(\omega) + ik_0\mathbf{T}_{car}(\omega)$, where $k_0 = \omega/c$ is the wavenumber in vacuum. Once the electric and toroidal Cartesian dipole moments are equal in magnitude and show opposite phases, $\mathbf{P}_{car}(\omega) = -ik_0\mathbf{T}_{car}(\omega)$, the spherical electric dipole moment $\mathbf{P}_{sph}(\omega)$ vanishes, suppressing the radiated field. This situation describes the condition for the excitation of an optical anapole[5]. In Supplementary Note 4, we provide further details on the calculation of the spherical and Cartesian multipoles as well as on the multipole expansion in the long-wavelength approximation.

To compare the multipole decomposition with the EEL spectrum of the 250 nm disk, we calculate the scattered power from the dipole and quadrupole moments of the induced current density $\mathbf{J}_{ind}(\mathbf{r})$ (see Supplementary Note 4). The scattered power from these multipole orders adequately captures the spectral features of the EEL spectrum across the energy range from 0.5 eV to 1.5 eV. Above 1.5 eV the EEL spectrum reveals additional spectral features that can be reproduced by also considering the scattered power from higher-order multipoles.

Figure 1d shows the calculated scattered powers from the spherical electric dipole moment $P_{scat}^{\mathbf{P}_{sph}} \propto |\mathbf{P}_{sph}(\omega)|^2$ (black dashed line), as well as the Cartesian electric $P_{scat}^{\mathbf{P}_{car}} \propto |\mathbf{P}_{car}(\omega)|^2$ (green line) and toroidal $P_{scat}^{\mathbf{T}_{car}} \propto |ik_0\mathbf{T}_{car}(\omega)|^2$ (magenta) dipole moments. The spectrum of $|\mathbf{P}_{sph}(\omega)|^2$ features a dip at around 1.25 eV marked as $A_{11}^E$. A direct comparison of the EEL spectrum (Fig. 1c) and the spectrum of the scattered power by the spherical dipole moment (black dashed line in Fig. 1d) shows that the dip in $|\mathbf{P}_{sph}(\omega)|^2$ matches the lowest-energy dip in $\Gamma(\omega)$, indicated by the gray dashed lines. In addition, this dip occurs at the energy where the electric and toroidal Cartesian dipole moments induced in the nanodisk show equal magnitude and opposite phase (intersection of green and magenta lines at $A_{11}^E$ in Fig. 1d). This confirms that the lowest-energy dip in the EEL spectra of the nanodisk

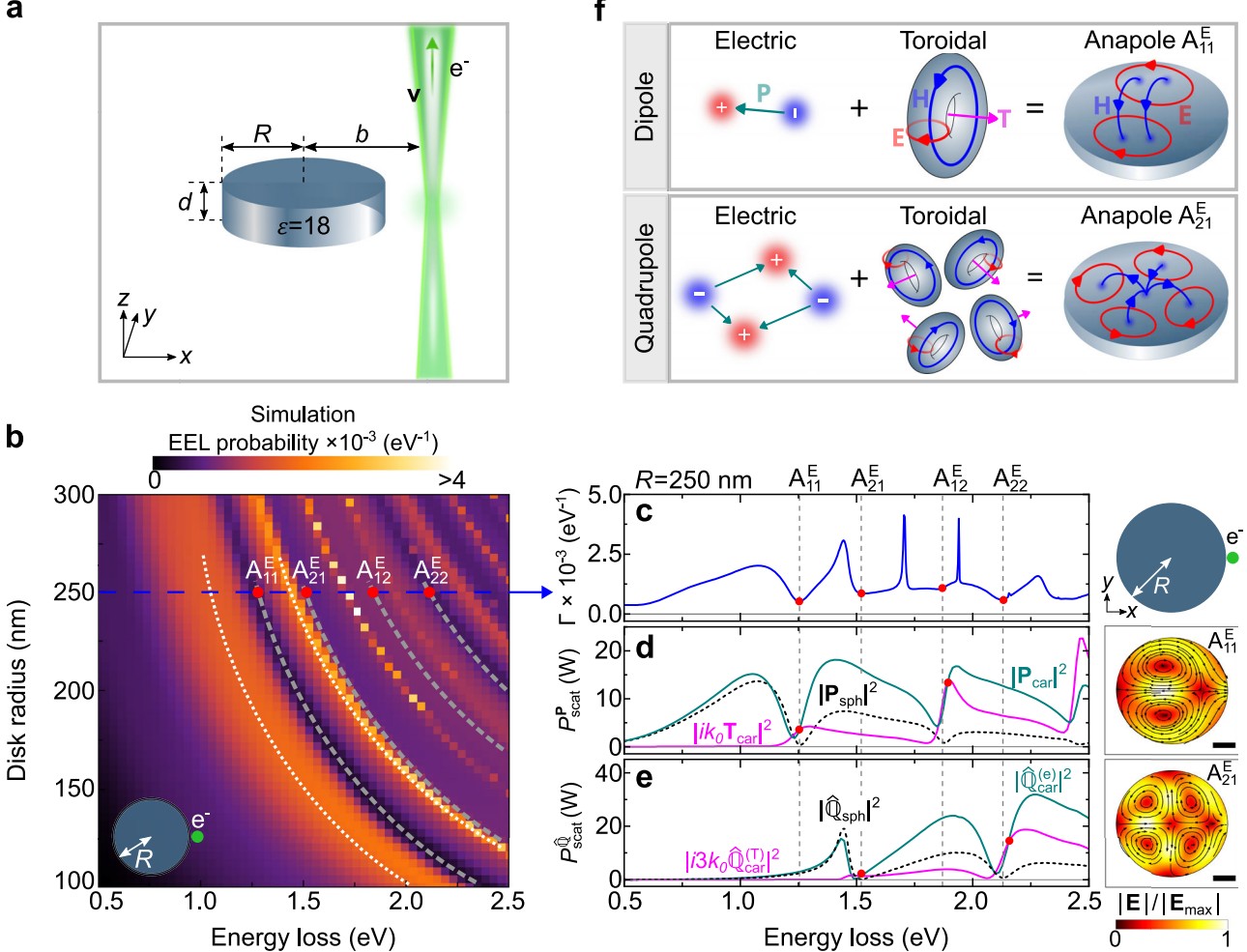

**Fig. 1 | Theoretical description of optical anapoles excited by fast electron beams. a** Sketch of the system under study: a high-index dielectric nanodisk (blue cylinder) with dielectric function $\varepsilon = 18$, thickness $d = 55$ nm and radius $R$ excited by a focused electron beam (green ray, $e$ is the elementary charge) traveling along the $z$-direction with velocity $v$ at a distance $b$ (impact parameter) with respect to the nanodisk center. **b** EEL probability $\Gamma(\omega)$ calculated as a function of both the nanodisk radius $R$ and the energy loss experienced by the electron beam. White dotted lines are guides to the eye and mark the dispersion of the first two peaks in the EEL spectra. Gray dashed lines mark the dispersion of four dips in the EEL spectra and are labeled as $A_{ij}^E$, according to their symmetry as described in the text. **c** Simulated EEL probability spectrum (blue line) of a nanodisk with

$R = 250$ nm. The red dots mark the spectral dip positions. **d** Scattered power from the $\mathbf{P}_{sph}$, $\mathbf{P}_{car}$ and $\mathbf{T}_{car}$ dipole moments induced in the nanodisk with $R = 250$ nm. **e** Scattered power from the $\hat{\mathbb{Q}}_{sph}$, $\hat{\mathbb{Q}}_{car}^{(e)}$ and $\hat{\mathbb{Q}}_{car}^{(T)}$ quadrupole moments induced in the nanodisk with $R = 250$ nm. The field plots on the right of panels **d** and **e** depict the amplitude of the total electric field $|\mathbf{E}(\omega)|$ inside the disk at the plane $z = 0$ (half of the nanodisk height), for energies: ($A_{11}^E$) 1.255 eV and ($A_{21}^E$) 1.52 eV. The scale bar is 100 nm. The field plots are normalized to the maximum value $|\mathbf{E}_{max}|$ in each case (from top to bottom): $4.2 \times 10^8$ V/m and $4.0 \times 10^8$ V/m. The inset next to panel **c** illustrates a top view of the nanodisk probed by the electron beam. **f** Sketch of the anapole states formed by the electric and toroidal dipoles and by the electric and toroidal quadrupoles.

is associated with the optical anapole excited by the fast electron beam.

The plot of the spatial electric field distribution at 1.255 eV inside the disk (right panel of Fig. 1d) shows the typical anapole pattern with regions of intense fields inside the nanodisk and two field vortices. We refer to this anapole as the first electric dipole, $A_{11}^E$, anapole state. We note that the ideal anapole (zero in the EEL probability) is not fully achieved due to higher-order multipolar contributions to the energy loss, and thus, an "attenuated dip" can be observed in the EEL spectra.

Analogously, we can understand the dip at 1.52 eV in the EEL spectrum (marked as $A_{21}^E$ in Fig. 1c) as due to the destructive interference between electric and toroidal quadrupole current distributions with identical far-field patterns[40]. To demonstrate this, we calculate the electric spherical $\hat{\mathbb{Q}}_{sph}(\omega)$, Cartesian electric $\hat{\mathbb{Q}}_{car}^{(e)}(\omega)$ and toroidal $\hat{\mathbb{Q}}_{car}^{(T)}(\omega)$ quadrupole moments of the induced current density $\mathbf{J}_{ind}(\mathbf{r})$. In the long-wavelength approximation, one finds that the spherical electric quadrupole moment has the following form

$\hat{\mathbb{Q}}_{sph}(\omega) \approx \hat{\mathbb{Q}}_{car}^{(e)}(\omega) + 3ik_0 \hat{\mathbb{Q}}_{car}^{(T)}(\omega)$ (see Supplementary Note 4) and thus, the condition for the excitation of electric quadrupole anapole states is $\hat{\mathbb{Q}}_{car}^{(e)} = -3ik_0 \hat{\mathbb{Q}}_{car}^{(T)}$.

In Fig. 1e we show the scattered power from the spherical electric quadrupole moment $P_{scat}^{\hat{\mathbb{Q}}_{sph}} \propto |\hat{\mathbb{Q}}_{sph}(\omega)|^2$ (black dashed line), Cartesian electric $P_{scat}^{\hat{\mathbb{Q}}_{car}} \propto |\hat{\mathbb{Q}}_{car}^{(e)}(\omega)|^2$ (green line) and toroidal $P_{scat}^{\hat{\mathbb{Q}}_{car}} \propto |3ik_0 \hat{\mathbb{Q}}_{car}^{(T)}(\omega)|^2$ (magenta line) quadrupole moments. From the green and magenta spectra we clearly see that the Cartesian and toroidal quadrupole moments have the same amplitude but opposite phase at 1.52 eV, which confirms that the dip at 1.52 eV in the EEL spectra corresponds to the excitation of the first electric quadrupole anapole state labeled as $A_{21}^E$. The difference in the properties of the first electric dipole and the first electric quadrupole anapoles excited by the fast electron beam can be observed in the right panel of Fig. 1e, where we show the spatial field distribution induced at 1.52 eV (marked with $A_{21}^E$)

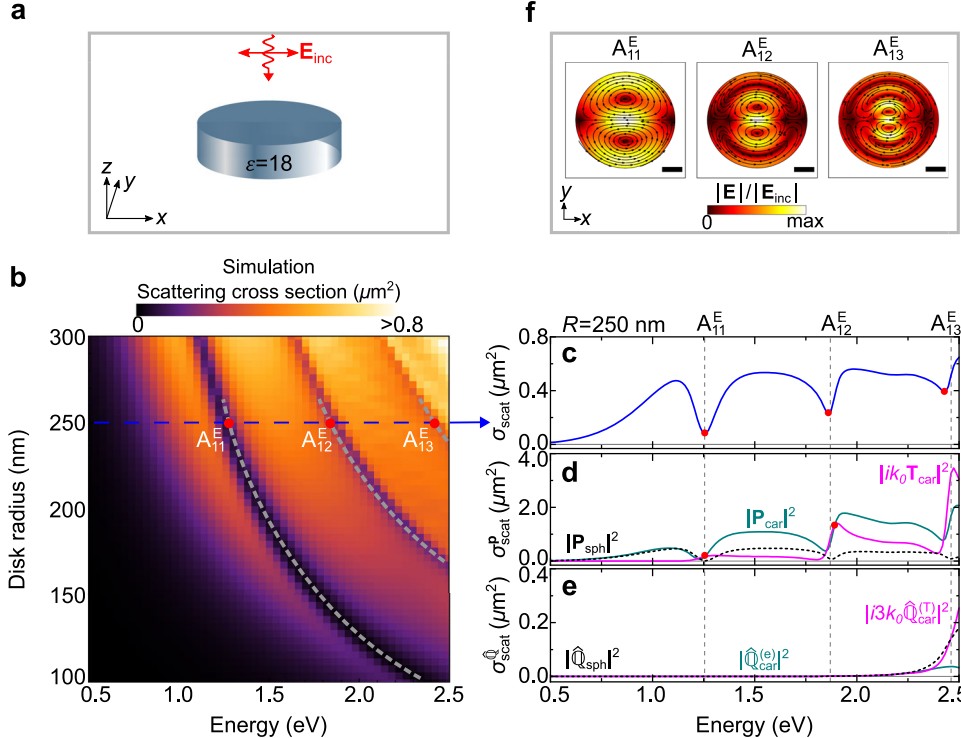

**Fig. 2 | Optical anapoles excited by plane wave illumination. a** A high-index dielectric nanodisk (blue cylinder) is excited by a plane wave ($\mathbf{E}_{inc}$, red arrow) that propagates perpendicular to the top surface of the disk (normal incidence) and is polarized along the *x*-axis. **b** Simulated scattering cross section $\sigma_{scat}(\omega)$ of the nanodisk calculated as a function of both the disk radius *R* and the photon energy $\hbar\omega$. Gray dashed lines are guides to the eye and indicate the dispersion of the first three dips in the scattering cross section spectra. **c** $\sigma_{scat}(\omega)$ obtained for the nanodisk with $R = 250$ nm. The red dots mark the spectral dip positions. **d** Partial

scattering cross section of the $\mathbf{P}_{sph}$, $\mathbf{P}_{car}$ and $\mathbf{T}_{car}$ dipole moments induced in the nanodisk with $R = 250$ nm. **e** Partial scattering cross section of the $\hat{\mathbf{Q}}_{sph}$, $\hat{\mathbf{Q}}_{car}^{(e)}$ and $\hat{\mathbf{Q}}_{car}^{(T)}$ quadrupole moments induced in the nanodisk with $R = 250$ nm. **f** Amplitude of the total electric field $|\mathbf{E}(\omega)|$ inside the disk at the plane $z = 0$, for energies: ($A_{11}^E$) 1.255 eV, ($A_{12}^E$) 1.87 eV and ($A_{31}^E$) 2.46 eV. The scale bar is 100 nm. The field plots are normalized to the amplitude of the incident plane wave $|\mathbf{E}_{inc}|$. In each case, the maximum value of $|\mathbf{E}(\omega)|/|\mathbf{E}_{inc}|$ is (from left to right): 3.0, 5.4 and 4.7.

inside the disk. Notice that the field distribution of the $A_{21}^E$ anapole exhibits two additional nodes compared to the field distribution of the $A_{11}^E$ anapole (see right panel of Fig. 1d). This increase in the number of nodes can be attributed to the quadrupolar distribution of the induced current density $\mathbf{J}_{ind}(\mathbf{r})$ inside the disk, as sketched in Fig. 1f.

The subsequent dips in the EEL spectrum (marked as $A_{12}^E$ and $A_{22}^E$ in Fig. 1c) are associated with the excitation of higher-order anapole states such as the second electric dipole anapole (marked as $A_{12}^E$) and the second electric quadrupole anapole (marked as $A_{22}^E$) states. These dips originate from the destructive interference of the radiation produced by Cartesian electric and higher-order toroidal multipoles excited in the nanodisk[40]. These higher-order multipoles are connected with additional higher-order terms, and can be also referred to as the mean-square radii of the respective multipoles[41]. The contribution of these higher-order multipoles is larger at higher energies (shorter wavelengths), and thus the conventional Cartesian multipole decomposition in the long-wavelength approximation fails to describe the spectral positions of $A_{12}^E$ and $A_{22}^E$ (compare the position of the red dots and the gray dashed lines above 1.75 eV in Fig. 1d, e). The spherical multipole decomposition, on the other hand, accurately reproduces the dips (anapole states) appearing in the EEL spectra. This allows us to label the *j*-th dip of the scattered power from the electric spherical $2^i$-pole as $A_{ij}^E$.

To show the advantage of EELS over far-field optical spectroscopy in probing anapoles, we compare the EEL spectra of the high-index dielectric disks with the far-field optical scattering spectra of the same disks. To that end, we calculate the scattering cross section $\sigma_{scat}(\omega)$ of the disks shown in Fig. 1a illuminated by a linearly-polarized plane wave

propagating along the *z*-axis (illustrated in Fig. 2a). The result is shown in Fig. 2b, c (see "Methods" and Supplementary Note 1 for the details of the numerical simulations). Similar to the EEL spectra $\Gamma(\omega)$, we observe dips in $\sigma_{scat}(\omega)$ that shift to higher energies as the nanodisk radius *R* decreases (see gray dashed lines in Fig. 2b). By performing a multipole decomposition of the current density $\mathbf{J}_{ind}(\omega)$ induced in the disk of $R = 250$ nm (see Fig. 2d, e), we can associate these dips (analog to the discussion of Fig. 1) with the excitation of the first ($A_{11}^E$), second ($A_{12}^E$) and third ($A_{13}^E$) electric dipole anapole states. The differences between these anapoles can be observed by plotting their spatial field distributions, as shown in Fig. 2f. We observe that the $A_{11}^E$ anapole exhibits the characteristic field distribution of the first electric dipole anapole state. The $A_{12}^E$ and $A_{13}^E$ anapoles, however, exhibit additional nodes at the edges of the disk, originating from resonant modes with higher radial order, i.e., higher number of nodes along the radial direction. On the other hand, quadrupolar contributions are negligible in the scattering cross section spectra, as compared to dipolar contributions (Fig. 2d and e), highlighting the advantage of EELS for probing higher-order anapole states.

The possibility to probe optical anapole states with fast electron beams turns EELS into a promising tool for fundamental studies of optical phenomena involving anapoles. To experimentally verify our numerical calculations, we fabricate high-index TMDC WS₂ nanodisks with various radii and performed electron energy loss spectroscopy to spatially resolve their optical behavior.

## Fabrication of WS₂ nanodisks

We fabricate WS₂ nanodisks by transferring a mechanically exfoliated WS₂ flake onto a 50 nm thick SiN membrane and by performing a

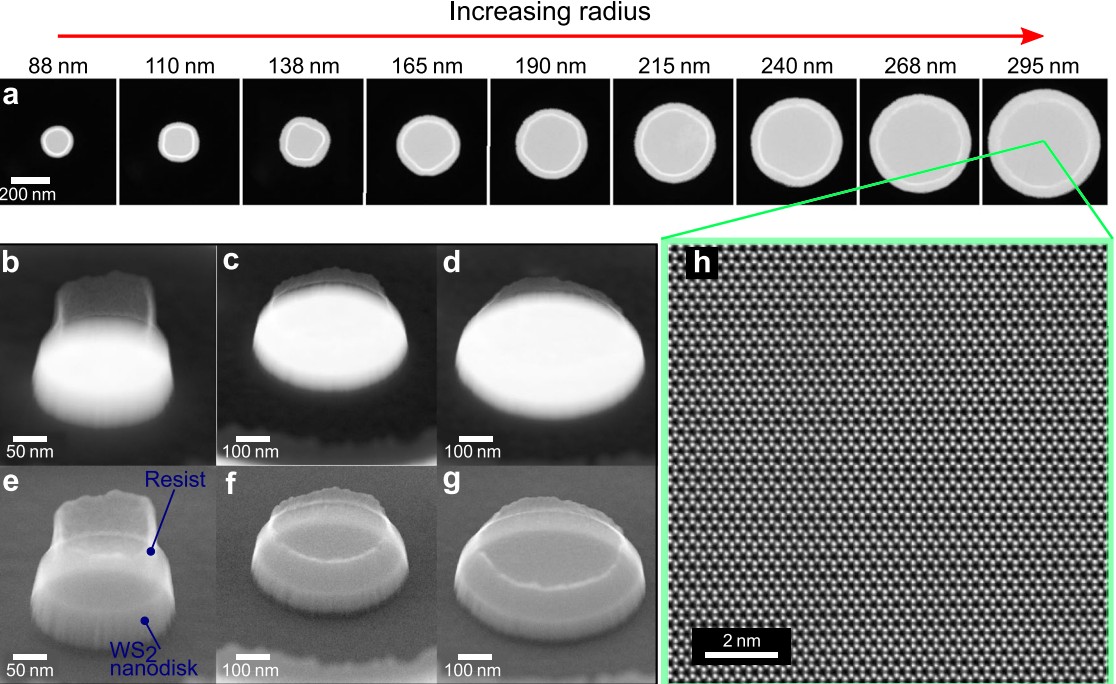

**Fig. 3 | Fabrication of WS₂ nanodisks. a** Plan view HAADF STEM images of nine WS₂ nanodisks with radii ranging from 88 nm to 295 nm. Disk radii are shown at the top of each image and a scale bar of 200 nm stands for all the images. **b–d** 53° tilted HAADF STEM images of the nanodisks with radii 110 nm, 240 nm and 295 nm, respectively. **e–g** 53° tilted SE STEM images simultaneously acquired of the three nanodisks shown in panels **b–d**. **h** Atomic-resolution HAADF STEM image taken from the center of the largest (295 nm) disk.

combination of e-beam lithography and dry etching. This process enables the creation of donut-shaped etched patterns with isolated nanodisks at their center with selected radii (see "Methods" and Supplementary Note 5 for additional details). The thickness of the nanodisk determines its optical response to a probing fast electron, therefore we experimentally measure the disk thickness using three independent methods: EELS, tilted-view STEM imaging, and the combination of optical reflectivity and transfer-matrix fitting (Supplementary Note 6). From the three methods, we find a disk thickness of around $d \approx 70$ nm.

In Fig. 3a we show high-angle annular dark-field (HAADF) STEM images of nine WS₂ nanodisks of different size, demonstrating the ability to precisely fabricate isolated nanodisks with controllable radii ranging from around 88 nm to 295 nm. Simultaneously acquired 53° tilted HAADF and secondary electron (SE) STEM images of nanodisks with various radii (Fig. 3b–g) reveal the morphology of the nanoresonators and the residual resist material that remains on top of the WS₂ nanodisks after fabrication, as indicated in Fig. 3e. High-angle annular dark-field STEM images are dominated by mass-thickness Z-contrast[42] and thus, the brightest regions reveal the position and shape of the WS₂ nanodisks below the primarily low-Z residual resist material.

To identify the size and morphology of both the WS₂ nanodisks and the residual material, we use SE image contrast which is sensitive to surface topography[43]. From Fig. 3b–g we observe two main features of the WS₂ nanodisks: (i) the edges show small vertically aligned surface variations and (ii) a tapered side surface with larger radii at the base. These variations of the WS₂ nanodisk radius are small compared to the average nanodisk radius, and thus modeling our nanodisks as perfect disks is an adequate description of the system. The residual resist could lead to minor shifts in energy of the disk modes, but does not alter the excitation of the nanoresonator modes and anapole states. Therefore, the resist has not been included in our model calculations. We address the reader to Supplementary Note 7, where we provide further details on the chemical composition of the fabricated samples. In Fig. 3h we

show an atomic-resolution HAADF STEM image from the center of a nanodisk, revealing that the single crystalline atomic structure of the WS₂ flake is preserved after the nanodisk fabrication process.

## Probing optical anapoles in WS₂ nanodisks using EELS

To experimentally investigate the optical response of WS₂ nanodisks to a fast electron beam, we perform monochromated STEM EELS experiments using a 200 keV electron beam with less than 1 nm spatial resolution and 20 meV − 40 meV energy resolution (see "Methods"). In Fig. 4a we show the collected EEL spectra as a function of nanodisk radius as obtained when an aloof electron beam passes outside ($b < R + 5$ nm) the edge of each nanodisk shown in Fig. 3a. From the EEL spectra we observe a low-energy loss signal composed of multiple sharp peaks and dips (see Fig. 4a). The number of peaks and dips decreases steadily as the disk size is reduced and their position shifts monotonously to higher energies in agreement with our theoretical prediction (Fig. 1b).

All collected EEL spectra of the WS₂ disks exhibit a spectral peak at around 1.95 eV (bright region in Fig. 4a), above which the EEL signal dampens and blurs. For a better quantitative comparison, we show in Figs. 4b, c individual experimental EEL spectra of the $R = 268$ nm (blue spectrum) and $R = 110$ nm (red spectrum) nanodisks, respectively. The peak at 1.96 eV is due to the excitation of the A-exciton of WS₂ (see Supplementary Note 1), as identified in the EEL spectrum of an unpatterned WS₂ flake shown by the green line in Figs. 4b, c.

To better understand the measured EEL spectra, we calculate numerically the EEL probability $\Gamma(\omega)$ for WS₂ nanodisks with similar radius ranging from 88 nm to 295 nm and thickness $d = 70$ nm (Fig. 4d). In contrast to the numerical simulations shown in Fig. 1, the calculations shown in Figs. 4d–f are performed with a frequency-dependent permittivity function that includes the A-exciton resonance of WS₂ (see Supplementary Note 1). The results of these simulations show a good agreement with the experimental spectra in the number, position, and dispersion of peaks and dips across the complete set of disk sizes

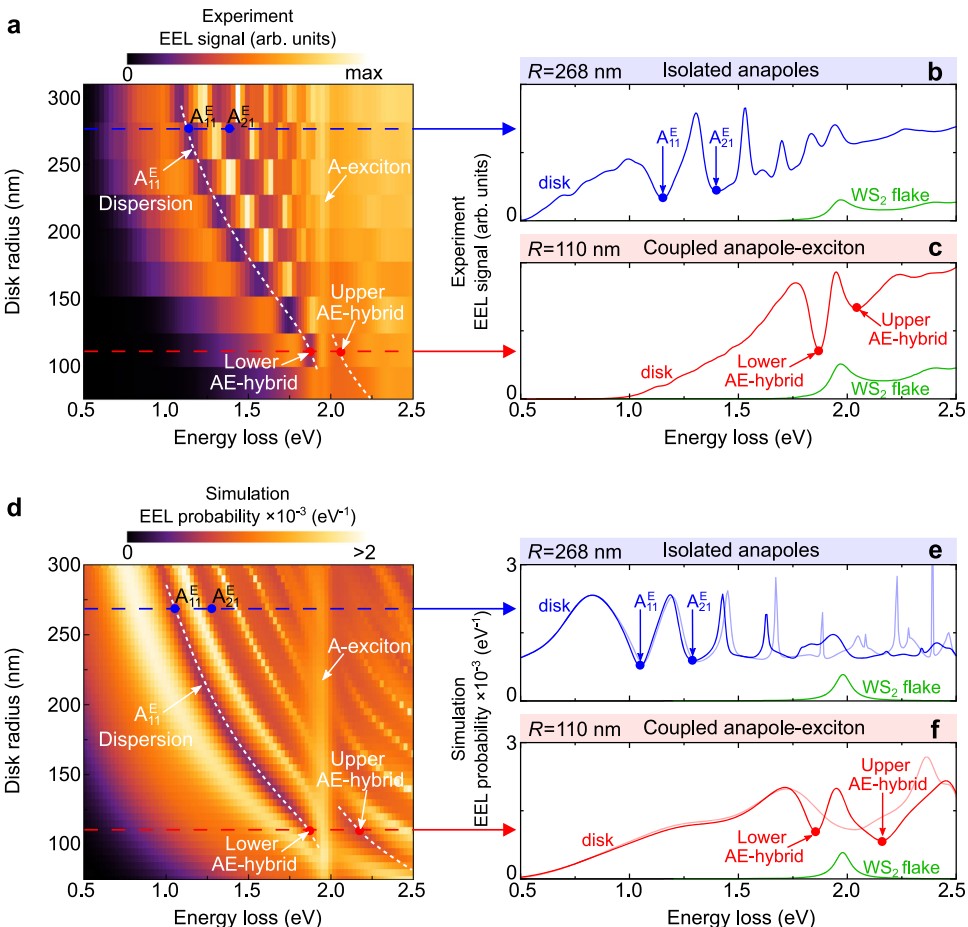

**Fig. 4 | Electron energy loss spectra of the WS₂ nanodisks. a** Experimental EEL spectra of the nine disks displayed in Fig. 3a. For better visualization, we show in Supplementary Note 8 the experimental EEL spectra for each disk radius. Blue solid line in **b** and red solid line in **c** show the EEL spectra for disks with $R = 268$ nm and $R = 110$ nm, respectively. Green lines in panels **b** and **c** show experimental EEL spectra from an unpatterned WS₂ flake. **d** Simulated EEL probability $\Gamma(\omega)$ as a function of both the nanodisk radius $R$ and the energy loss experienced by the electron beam. Blue solid line in **e** and red solid line in **f** show the simulated EEL spectra for disks with $R = 268$ nm and $R = 110$ nm, respectively. Thin blue and thin red spectra correspond to $\Gamma(\omega)$ obtained with disks of artificial permittivity $\varepsilon = 18$. Green lines in panels **e** and **f** show calculated $\Gamma(\omega)$ for a WS₂ flake of 70 nm thickness (see Supplementary Note 1). White dashed lines in panels **a** and **d** are guides to the eye indicating anti-crossing of the $A_{11}^{E}$ anapole and the A-exciton. Green curves in **b** and **c** are scaled to be consistent with the relative heights in **e** and **f**.

(compare Figs. 4a, d). The excitation of the A-exciton can be consistently observed both in the experimental and simulated spectra. Above 2 eV, the peaks and dips in the simulated spectra are better resolved than in the experimental spectra, which is due to the excitation of B- and C-exciton resonances in WS₂ that are not included in the numerical simulations.

To check whether the dips in the experimental spectra are due to the excitation of optical anapoles, we analyze the EEL signal of the 268 nm radius nanodisk (blue lines in Fig. 4b, e). Both the experimental and simulated EEL spectra show two attenuated dips between 1.0 eV and 1.5 eV, marked with $A_{11}^{E}$ and $A_{21}^{E}$ in Figs. 4b, e. As discussed above in Fig. 1, these two dips can be associated with the first electric dipole and the first electric quadrupole anapole states excited in the nanodisk. To verify that the dips in this energy range are not caused by material losses of WS₂, we simulate the EEL spectrum of the model disk with $R = 268$ nm, $d = 70$ nm and artificial permittivity $\varepsilon = 18$ mimicking that of WS₂ without the A-exciton resonance (thin blue line in Fig. 4e). By comparing the solid and thin blue lines, we can observe that the two lowest-energy dips (marked $A_{11}^{E}$ and $A_{21}^{E}$) appear nearly at the same energies in both spectra, which confirms that the dips are not caused by the material losses of WS₂. The differences between the solid and the thin blue spectra in Fig. 4e are a direct consequence of the appearance of the A-exciton resonance at 1.96 eV (identified in the EEL spectrum of a WS₂ flake shown by the green line in Fig. 4e). We note

that multilayer WS₂ is a natural anisotropic material; however, the anisotropy in this case does not influence the anapoles excitation, and the spectral response of the disk with isotropic permittivity is nearly identical to that of an anisotropic disk (see Supplementary Note 9). In addition, we confirm that the reflectivity of WS₂, within the energy range of 1.0 eV to 1.6 eV, is at least 2.5 times higher than the reflectivity of SiN. This allows us to discard substrate effects and also corroborates that the spectral features in the EEL spectra are predominantly due to optical excitations of the WS₂ disk.

**Anapole-exciton hybridization.** The coexistence of an excitonic resonance and the dispersive anapoles in the same object allows these resonant features to couple and hybridize with each other with different levels of strength as the nanodisk radius $R$ is varied. To explore this aspect, we trace the $A_{11}^{E}$ anapole state upon decreasing the nanodisk radius from 268 nm to 110 nm (white dashed line in Figs. 4a, d) until it reaches the energy of the A-exciton resonance, where a splitting (anti-crossing) of the dip is produced. This behavior is clearly shown in Fig. 4f, where we plot the simulated spectrum of the 110 nm disk radius (solid red line, extracted from Fig. 4d). For comparison, we also plot the spectrum of the model disk with artificial permittivity $\varepsilon = 18$ (thin red line). By comparing the spectra of both types of disks, one can observe that the attenuated anapole dip at 2 eV (thin red line) splits into two dips at 1.86 eV and 2.16 eV in the solid red line. Due to the

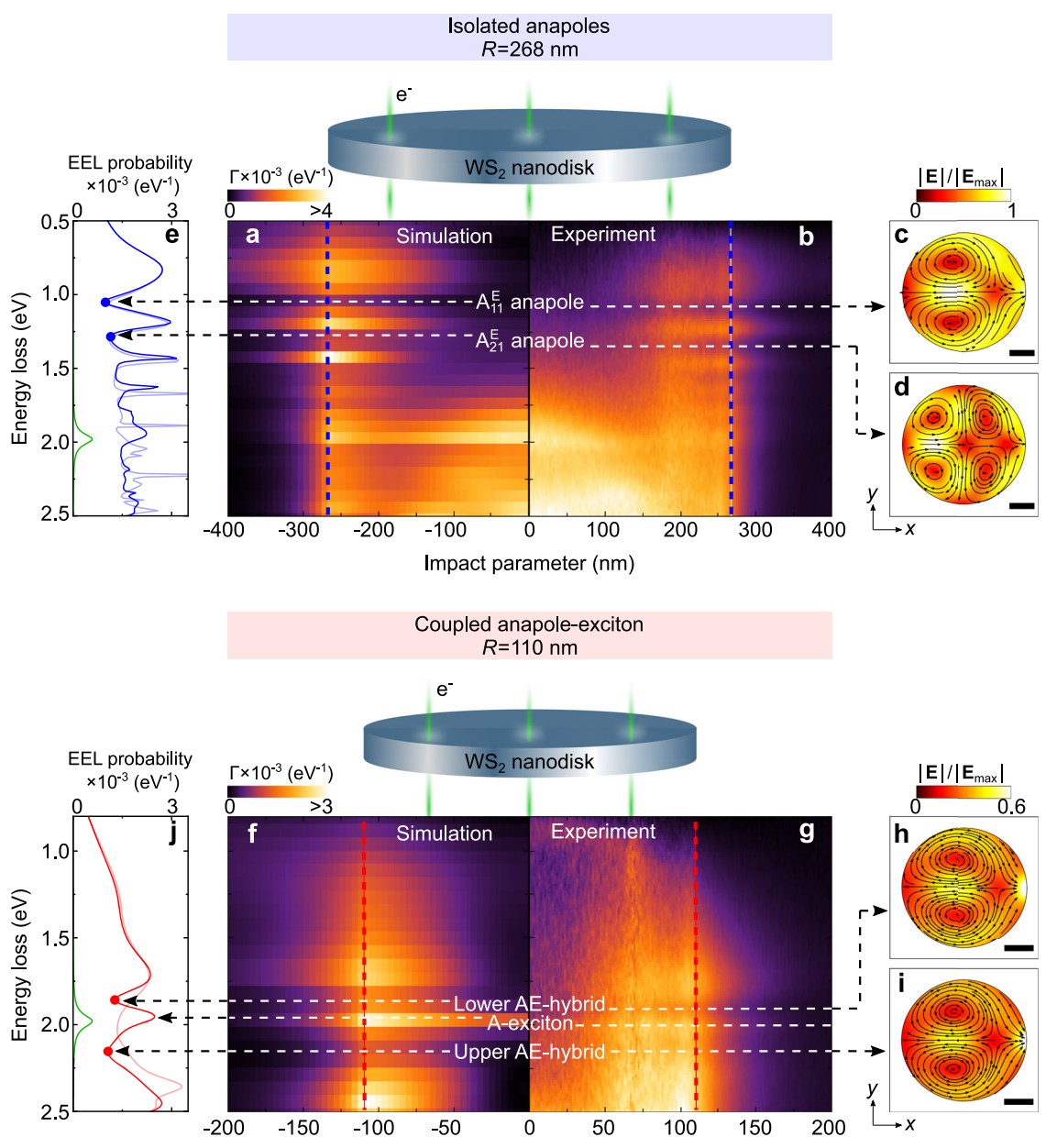

**Fig. 5 | Spatially-resolved EEL of WS$_2$ nanodisks. a** Simulated and **b** experimental EEL spectra recorded along the impact parameter $b$ of a disk with 268 nm radius as depicted in the schematics above the spectra. Blue dashed lines indicate $b = 268$ nm. White dashed lines indicate the anapoles A$_{11}^E$ (at 1.048 eV) and A$_{21}^E$ (at 1.276 eV). Panels at the right of **b** show the amplitude of the total electric field $|\mathbf{E}(\omega)|$ at the plane $z = 0$ for the energies **c** 1.048 eV and **d** 1.276 eV. The scale bar is 100 nm and the field plots are normalized to the maximum value $|\mathbf{E}_{max}|$ in each case: **c** $3.2 \times 10^8$ V/m and **d** $3.4 \times 10^8$ V/m. **e** Simulated $\Gamma(\omega)$ spectra for the WS$_2$ disk (solid blue line), the model disk with $\varepsilon = 18$ (thin blue line) and a WS$_2$ flake of 70 nm thickness (green line). **f–j** same as **a–e** but for the $R = 110$ nm nanodisk. The scale bars in panels **h** and **i** are 50 nm. The maximum value $|\mathbf{E}_{max}|$ in each case is: **h** $1.2 \times 10^9$ V/m and **i** $1.4 \times 10^9$ V/m.

hybrid nature of these dips, we refer to them as the lower anapole-exciton-hybrid (lower AE-hybrid) and the upper anapole-exciton-hybrid (upper AE-hybrid). We can also see a peak in between the two dips which originates from the excitation of excitons that do not couple to the anapole. The splitting of the dips, together with the anti-crossing feature, are signatures of the coupling between the A$_{11}^E$ ana-pole state and the A-exciton, consistent with previous observations in far-field optical spectroscopy of WS$_2$ nanodisks[35].

The anti-crossing observed, and fully identified in Fig. 4d resembles the typical situation of coupling between an EM mode confined in an optical cavity and a dipolar excitation. The ana-pole, however, is not an EM mode of the disk, but instead is the

result of interference between at least two resonant modes in the disk excited by the electron beam[44,45]. To explain the coupling between the A$_{11}^E$ anapole and the A-exciton, we implement an analytical model of the response of the coupled anapole-exciton system based on temporal coupled-mode theory (TCMT). Within this framework, we model the anapole-exciton system using an effective $3 \times 3$ Hamiltonian containing the eigenfrequencies of two EM modes, whose far-field interference produces the anapole dip, coupled to a third non-radiating mode representing the A-exciton of WS$_2$.

We use this model and apply the following procedure to repro-duce the experimental and simulated EEL spectra. First, we reproduce

the simulated spectra of the model disk with $\varepsilon = 18$ and find the eigenfrequencies of the two EM modes producing the $A_{11}^E$ anapole dip. We then use these values to reproduce the experimental and simulated EEL spectra of the $WS_2$ disks (Figs. 4a, d) and find the coupling strengths between each EM mode and the A-exciton resonance. By diagonalizing the effective $3 \times 3$ Hamiltonian of the system, we find the eigenfrequencies of the new hybrid modes. The results obtained with this procedure are shown in Supplementary Note 10, where we describe in detail the coupled-mode model. The analysis reveals a clear anti-crossing between the hybrid modes as one varies the nanodisk radius, indicating that the two electromagnetic modes are strongly coupled to the A-exciton resonance. The two dips that result from the coupling between the two modes and the A-exciton resonance correspond to the lower and upper AE-hybrids.

### Real-space mapping of optical anapole states

An advantage of EELS in STEM is its ability to acquire spectral information of a sample with subnanometer spatial resolution. Typically, this is achieved by scanning the sample area with the fast electron beam, thus obtaining spectral information of the sample at different beam positions (see Methods). We apply this capability to spatially resolve the resonant modes and anapoles states excited in the $WS_2$ nanodisk. To that end, we collect the experimental EEL signal as a function of the position of the fast electron with respect to the nanodisk center (impact parameter $b$). The cylindrical symmetry of the nanodisk along the $z$-axis, together with the trajectory of the probing electron beam, yields an EEL signal that depends only on the impact parameter. This allows us to present the spatial distribution of the EEL signals obtained from the $WS_2$ nanodisks by a 2D-EEL line profile showing the energy loss as a function of impact parameters for each energy.

Figure 5 shows experimental and simulated 2D-EEL line profiles for $R = 268$ nm (Figs. 5a, b) and $R = 110$ nm nanodisks (Figs. 5f, g). The line profiles for the $R = 268$ nm nanodisk reveal a low-loss EEL signal confined to an annular region with a maximum at the edge of the nanodisk (see region between 0.5 eV to 1.5 eV in Figs. 5a, b). We show in Fig. 5e the simulated spectrum for a beam that is close to the nanodisk edge (blue dashed line in Fig. 5a). We recognize the $A_{11}^E$ and the $A_{21}^E$ anapole dips at around 1 eV and 1.25 eV, respectively. The electric field distribution inside the nanodisk at these energies (see Figs. 5c, d) corroborate the nature of these dips. Interestingly, when the electron beam passes through the nanodisk center ($b = 0$) the EEL signal becomes nearly zero, as shown in Figs. 5a, b. In this case, the electron beam is not able to efficiently excite the disk modes due to the cylindrical symmetry of the disk, and thus the optical anapoles are also not excited by the electron beam.

Finally, the line profile obtained from the $R = 110$ nm nanodisk reveals an EEL signal that is spatially confined to an annular region from half the disk radius to significantly outside of the nanodisk. The calculated electric field distributions at the spectral dips (around 1.75 eV and 2.25 eV as indicated in Fig. 5j) display a clear electric dipole anapole-like field pattern (Figs. 5h, i). These field distributions corroborate that the first electric dipole anapole is hybridized with the A-exciton (bright region around 1.95 eV) to produce the anapole-exciton hybrids. These results open up the possibility to explore, in future EELS experiments, systems exhibiting more complicated spatial behavior of the isolated anapoles or anapole-exciton hybrids by, for example, breaking the cylindrical symmetry of the system.

## Discussion

In summary, we demonstrated that electron energy loss spectroscopy can be applied to probe optical anapole states in high-index dielectric nanoresonators. To that end, we calculated the electron energy loss probability of high-index dielectric nanodisks and showed that the prominent dips in the EEL spectra are associated with the excitation of optical anapoles in the disk. We experimentally verified our theoretical

prediction by performing EELS of $WS_2$ nanodisks and revealed optical anapoles and anapole-exciton hybrid excitation within the same nanoobject. Additionally, we demonstrated that EELS in STEM allows for spatial mapping of $WS_2$ nanodisk modes, isolated anapoles and anapole-exciton hybrids with subnanometer resolution. By placing the electron beam at specific positions along the $WS_2$ nanodisk, we can effectively control the modes excitation and thus the formation of the optical anapole.

Our results show that EELS in STEM is a powerful tool to study dark scattering states and their complex interactions with the electronic structure of dielectric materials beyond the possibilities offered by conventional optical techniques. We anticipate that our results will enable new possibilities for studying higher-order and magnetic anapole states in dielectric nanoresonators with subnanometer spatial resolution.

## Methods

### Sample fabrication

The multilayer $WS_2$ (~70 nm) flake was mechanically exfoliated from bulk crystal (HQ graphene) onto polydimethylsiloxane (PDMS) stamps using the scotch-tape method, and then transferred onto a 50 nm thick silicon nitride (SiN) membrane TEM grid (TEMwindows.com) with an all dry-transfer method. The fabrication of $WS_2$ nanodisks was achieved using our previously developed method with a combination of e-beam exposure of a positive resist and dry etching[35,46,47]. In brief, nanopatterning $WS_2$ (~70 nm) flake into the nanodisks was carried out by first spin coating a positive ARP 6200.13 resist at 4000 rpm for 1 min, followed by soft-baking at 120 °C for 5 min. To fabricate the nanodisks on a thin SiN membrane by dry etching, the sample was exposed with donut-shaped patterns with various inner and outer diameters using a Raith EPG5200 electron beam lithography system (100 keV) to make a resist mask for further dry etching (see Supplementary Note 5). It is worth mentioning that the inner diameter of the donut pattern defines the diameters of the nanodisks. Then, the sample was developed with $n$-amyl acetate for 4 min and dried gently with nitrogen gas. Subsequently, the sample was etched using a dry reactive ion etcher (RIE) with $CHF_3$ gas. A complete etching of $WS_2$ nanodisks was ensured by performing STEM diffraction, atomic-resolution STEM imaging, EELS and energy dispersive spectroscopy (EDS) chemical analysis inside the etched donut-shaped patterns.

### Experimental STEM imaging and EELS

STEM experiments were performed at 200 keV and at room temperature on a JEOL Mono NEO ARM 200F instrument. The microscope is equipped with a Schottky field emission gun, double Wien filter monochromator, probe aberration corrector, image aberration corrector, and Gatan Imaging Filter continuum HR spectrometer. HAADF and SE STEM images were acquired with an around 0.1 nm electron probe. The STEM image shown in Fig. 3h was produced by summing a series of 50 images after the series was registered based on cross correlation to correct for rigid image drifts between frames. The resulting summed image was Fourier filtered to further enhance signal-to-noise ratio and the $WS_2$ signal. EDS experiments utilized a double detector system with a collection solid angle of up to 1.6 sr.

EELS experiments were performed with less than 1 nm spatial resolution, 20 meV – 40 meV energy resolution, a 30 pA – 50 pA beam current, a 21 mrad electron probe convergence angle, and a 10 mrad EELS collection angle. The EEL spectra shown in Fig. 4a–c, were acquired from just outside the edge of the $WS_2$ nanodisks and are the sum of a series composed of 160, 000 spectra using a 5 meV dispersion. Before summing the spectral series, a high quality dark spectrum reference was subtracted from each spectra and the spectra were energy-aligned based on the zero energy loss peak position. After summing, the spectra were processed with the Richardson-Lucy deconvolution algorithm using a reference

spectrum acquired from the SiN window and only 5 iterations to ensure no introduction of known artifacts[48,49]. Deconvolution was used to reduce intensity from the zero-loss peak tail in the energy region of interest. Then the remaining tail of the zero-loss peak was removed by fitting each spectrum in the 0.3 eV – 0.5 eV range to a power law function, extrapolating that function and subtracting it from the spectrum.

The experimental EEL line profiles shown in Fig. 5 were each produced from three spectrum images of the same sample area. Each spectrum image consisted of $50 \times 3500$ spatial pixels and 2048 energy channels. To enhance signal-to-noise ratio, each spectrum image was binned by 50 in the $x$-direction and by 7 in the $y$-direction to produce a line profile, and the three profiles from each area were summed after the data processing mentioned below. The spectra were processed with high quality dark spectrum reference subtraction, energy-alignment, Richardson-Lucy deconvolution, and zero-loss peak tail removal using the same details as the spectra mentioned above. Additionally, the EEL line profiles were normalized based on their total integrated EEL signal. The EEL intensity in the as-acquired data is modified by scattering sources other than the low-loss optical and electronic signals that we are interested in. For example, $WS_2$ elastic diffraction scatters a significant amount of electrons outside of the spectrometer collection aperture, significantly reducing all the EEL signals when the electron beam passes through the $WS_2$ nanodisk as compared to when it passes outside the nanodisk. Additionally, variations in the morphology and composition of the residual resist modify the spatial variations in EEL signal intensity. Normalizing each spectrum in the line profile to its own total integrated EEL signal counteracts some of these effects and makes them more comparable to simulations. The spectrum images were acquired in dual EELS mode with a 15 meV dispersion to enable the collection of EELS data over a larger range of energies and increase the inelastic energy range of the normalization process. Splicing the two dual EEL spectra together allows each spectrum to extend to around 50 eV while still resolving the disk modes and anapole states in the 0.5 eV – 2.5 eV range.

### Numerical simulations

We performed the numerical simulations shown in Figs. 1, 2, 4 and 5 using the Radio Frequency Module of COMSOL Multiphysics software[50]. This module solves Maxwell's equations in the frequency domain based on the Finite Element Method (FEM). The nanodisk was modeled as a cylindrical structure of variable radius $R$ and thicknesses $d = 55$ nm (Figs. 1 and 2) and $d = 70$ nm (Figs. 4 and 5). The symmetry axis of the nanodisk is parallel to the $z$-direction and the center of the nanodisk was located at coordinates $(0, 0, d/2)$. The probing electron traveling in the $z$-direction was modeled as a line current (along the $z$-direction) located at a distance $b$ (impact parameter) with respect to the nanodisk center. For numerical calculations shown in Figs. 1 and 4, we located the electron beam trajectory outside the nanodisk at an impact parameter $b = 1.1 \times R$, whereas in Figs. 5e, j we used an impact parameter equal to $b = 1.05 \times R$. For simplicity, numerical calculations shown in Figs. 1 and 2 were performed without considering any substrate, whereas in Figs. 4 and 5 the nanodisk is on top of a 50 nm thick substrate layer characterized by the permittivity of SiN $\varepsilon_{SiN} = 4.1853$. We note that the SiN substrate does not alter the excitation of the anapole states. It slightly redshifts the resonant modes of the nanodisk, and thus, the anapole dips simply appear at lower energies (see Supplementary Note 11).

To ensure numerical convergence of the calculated electron energy loss probability $\Gamma(\omega)$, the complete structure (electron beam and nanodisk) was embedded in a homogeneous box filled with air of depth, width and height equal to $L_{PML} = 12 \times R$. We use perfectly matched layers (PML, with thickness equal to $0.1 \times L_{PML}$) for the boundaries of the simulation box, free triangular elements for the nanodisk mesh and free tetrahedral elements for all other structures. The length of the line current that models the electron beam is equal to $L_{PML}$. Material permittivities and further details on EEL probability and scattering cross section calculations are provided in Supplementary Note 1.

## Data availability

The authors declare that the data supporting the findings of this study are available within the paper and its supplementary information files or from the corresponding author(s) on reasonable request. Alternatively, the data can be accessed at https://digital.csic.es/.

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

## Acknowledgements

C.M.-E. thanks J. Olmos Trigo for fruitful discussions. C.M.-E. and J.A. further acknowledges grant no. IT 1526-22 from the Basque Government for consolidated groups of the Basque University and grants PID2019-107432GB-I00 and PID2022-139579NB-I00 both funded by MCIN/AEI/10.13039/501100011033 and by "ERDF A way of making Europe". R.H. acknowledges support from the Spanish Ministry of Science and Innovation under the María de Maeztu Units of Excellence Program (CEX2020-001038-M/MCIN/AEI/10.13039/501100011033) and Grant PID2021-123949OB-I00 funded by MCIN/AEI/10.13039/501100011033 and by "ERDF A way of making Europe". A.B.Y. acknowledges funding from the Swedish Research Council (VR, under grant No. 2020-04986). T.O.S. acknowledges funding from the Swedish Research Council (VR, Research environment grant No. 2016-06059). A.B.Y., B.M., E.O., and T.O.S. acknowledge funding from the Knut and Alice Wallenberg Foundation (KAW, under grant No. 2019.0140). D.G.B. acknowledges financial support from the Ministry of Science and Higher Education of the Russian Federation (Agreement No. 075-15-2021-606), Russian Science Foundation (grant No. 23-72-10005), and BASIS Foundation (grant No. 22-1-3-2-1). This work was performed in part at the Chalmers Material Analysis Laboratory (CMAL).

## Author contributions

C.M.-E., D.G.B., and T.O.S. conceived the idea. B.M. fabricated the samples under the supervision of T.O.S. A.B.Y. performed EELS measurements under the supervision of E.O. C.M.-E. performed numerical calculations under the supervision of R.H. and J.A. C.M.-E., D.G.B., J.A., and T.O.S. developed the theoretical model to describe the anapole-exciton system and analyzed the data. C.M.-E., A.B.Y., and D.G.B. wrote the manuscript with contributions from all authors. E.O., J.A., and T.O.S. supervised the project.

## Funding

## Competing interests

The authors declare no competing interests.
