## [Peer Review File · Nature Communications]

Reviewers' Comments:

Reviewer #1:

Remarks to the Author:

The authors investigated optical anapoles with an advanced technique so called EELS. They found that such a technique can map the anapoles excited in the WS₂ nanodisk with subnanometer resolution. Also, they found that excitation can be controlled by placing the electron beam at different positions on the nanodisks. Before I recommend it for publication, I have some concerns which shall be addressed

(1) The biggest concern is the novelty, I am not sure whether probing anapoles with fast electron beams is more than enough to justify the publication in NC. No doubt, EELS is a new technique and indeed provides some unique insights to re-examine the anapoles. However, as authors mentioned, anapoles has been intensively studied with near- and far-field optical techniques, the knowledge obtained from EELS seems not big surprise for me. Therefore, they need clarify the novelty of this work and highlight the advantage of EELS over near- and far-field probing techniques and point out the unique knowledge that cannot be obtained from aforementioned optical techniques, or may lead to some unique applications.

(2) Why the authors choose WS₂ nanodisks to study optical anapole, not silicon?

(3) In Fig.1b, the authors calculated EEL probability mapping and assign the dip as anapoles. I may suggest the authors also include the scattering cross section mapping to double confirm the existence of anapoles because it is the well-established way to define the anapoles.

(4) For Fig.3a, it is very hard to differentiate the strong coupling because avoided-crossing is not obvious. I may suggest authors presenting the EEL spectrum at different radius in the supplementary materials, which may help readers better observe strong coupling.

(5) In the experimental section, the authors define "impact parameter b" without any explanation, please add the description on it.

(6) Finally, some important references related to anapoles and TMDC bulks are missing.

(a) Application of anapoles in THG, L. Xu, et al, "Boosting third-harmonic generation by a mirror-enhanced anapole resonator", *Light: Science & Applications* 7,44(2018)

(b) High order anapoles, E. A. Gurvitz, et al "The High-Order Toroidal Moments and Anapole States in All-Dielectric Photonics", *Laser Photonics Reviews* 13(5),1800266(2019).

V. A. Zenin, et al, "Direct Amplitude-Phase Near-Field Observation of Higher-Order Anapole States", *ACS Nano* 17,7152(2017)

(c) TMDC Bulk, S. Busschaert, et al "Transition Metal Dichalcogenide Resonators for Second Harmonic Signal Enhancement", *ACS Photonics* 7, 2482(2020), L. Huang et al, "Enhanced light-matter interaction in two-dimensional transition metal dichalcogenides", *Rep.Prog. Phys* 85, 046401(2022)

Reviewer #2:

Remarks to the Author:

The authors study the excitation of anapole and hybrid anapole-exciton modes in WS₂ nanodisks by EELS and numerical calculations. They show that an e-beam propagating perpendicular to the plane of the disk excites non-radiating excitations in the disk arising from the interference of electric and toroidal multipoles. These excitations depend strongly on the position of the beam and vanish when the beam goes through the center of the disk. The authors also investigate the role of the disk radius and demonstrate hybrid anapole-exciton modes in small disks. By using coupled mode theory, they are able to retrieve the coupling strengths between the different modes of the structure.

The results presented in this work are intriguing and will be useful to the community. However, the paper is not well written and the results are not always clearly presented. In addition, certain key questions related to the validity of the results are not addressed. In particular:

- Figure 1 is misleading. Panel (a) shows the disk supported by a substrate, but the substrate is not taken into account in the calculations of panels (b-e).

- Also in Fig. 1: Panel (f) illustrates the formation of anapole modes arising from the interference

of electric dipole and toroidal dipole and from the interference of electric quadrupole and toroidal quadrupole. The resulting anapole modes are termed "First ED anapole" and "First QD anapole", which implies that that "Second ED anapole" is formed differently. The authors should clarify this point.

- It is difficult to compare the contributions of different multipoles as the units are different. The scattered power from each multipole should be used instead. See for instance doi: 10.1103/PhysRevE.65.046609.
- The authors take into account only four multipoles: electric dipole, toroidal dipole, electric quadrupole, and toroidal quadrupole. They should calculate all multipoles up to toroidal quadrupole order and provide evidence that these multipoles adequately represent the excitations of the disk.
- The reference list does not adequately represent the field. The term "anapole" was first introduced (in the static regime) by Zel'dovich, in J. Exp. Theor. Phys. 33, 1184–1186 (1957). The first experimental observation of a dynamic anapole was by Fedotov et al. in <https://doi.org/10.1038/srep02967>.
- How important is the anisotropy of the permittivity tensor in the excitation of anapole modes?
- The authors should discuss the dependence of the excited modes on the electron velocity.
- The samples studied experimentally include a thin substrate. Can the authors discuss the role of the substrate in the anapole excitations? For instance, are the electromagnetic fields of the resonant modes mainly confined in the disk or is there substantial excitation of the substrate too?
- In the Supplementary material, in Eq. (1) the limits of the integration refer to parameter L_{PML} , which is not defined.

To conclude, I would be happy to review a revised version of the paper, where said issues have been addressed.

Reviewer #3:

Remarks to the Author:

The authors have used Scanning Transmission Electron Microscopy (STEM) to probe optical anapoles in WS₂ nanodisks. Most characterization of anapole modes were conducted by using optical measurements so the use of electron energy loss spectroscopy seems to be novel. It may be useful for a small number of people but I don't believe it will reach a broad audience. The methodology is sound for both theoretical and experimental analyses. Also, the theoretical and experimental results have a good match.

Probing optical anapoles with fast electron beams

Response letter to the reviewer's report

Carlos Maciel-Escudero, Andrew B. Yankovich, Battulga Munkhbat, Denis G. Baranov, Rainer Hillenbrand, Eva Olsson, Javier Aizpurua, and Timur O. Shegai
(Dated: September 8, 2023)

We thank the reviewers for their comments that helped improve our manuscript. In the following, we reproduce in full the comments of the reviewers within the boxes. We provide our answer after each box, and the amendments made to the manuscript are highlighted in blue.

RESPONSE TO REVIEWER 1

General Assessment

The authors investigated optical anapoles with an advanced technique so called EELS. They found that such a technique can map the anapoles excited in the WS₂ nanodisk with subnanometer resolution. Also, they found that excitation can be controlled by placing the electron beam at different positions on the nanodisks. Before I recommend it for publication, I have some concerns which shall be addressed.

Response: We thank the reviewer for acknowledging our work. Below are our answers to the concerns of the reviewer.

Comment No. 1

The biggest concern is the novelty, I am not sure whether probing anapoles with fast electron beams is more than enough to justify the publication in NC. No doubt, EELS is a new technique and indeed provides some unique insights to re-examine the anapoles. However, as authors mentioned, anapoles has been intensively studied with near- and far-field optical techniques, the knowledge obtained from EELS seems not big surprise for me. Therefore, they need clarify the novelty of this work and highlight the advantage of EELS over near- and far-field probing techniques and point out the unique knowledge that cannot be obtained from aforementioned optical techniques, or may lead to some unique applications.

Response: We appreciate the comment of the reviewer which help us to be more precise on the advantages of EELS in this context. As compared to far-field optical spectroscopy, EELS has the advantage of accessing not only electric dipole modes in a resonator but also quadrupoles and higher-order modes that do not or weakly couple to standard far-field illumination. Therefore, we can access to optical modes that are not easily detectable by far-field optical spectroscopy. Even if we compare the information obtained with EELS as presented here with other localized probes such as those in near-field optical microscopy and spectroscopy, EELS probing of anapoles presents the following advantages:

- **Non-disturbing near-field probe:** EELS is a non-disturbing spectroscopy technique that does not modify the electromagnetic fields and modes of a nanoresonator. By positioning the electron beam on aloof trajectories, we can reduce the distortion produced by the probe (the localized electron) on the nanoresonator modes. This is a very well-known problem when using near-field optical techniques, which have been sometimes used to spatially map optical anapoles. For example, in Refs. “Miroshnichenko, *et al. Nat. Commun.* 6, 8069 (2015)” and “Zenin, *et al. Nano Lett.* 17, 11 (2017)”, the authors obtained spatial near-field images of anapoles excited in silicon nanodisks using aperture-based (‘Miroshnichenko, *et al.*) and apertureless (Zenin, *et al.*) scanning near-field optical microscopy (SNOM). The coupling between the near-field probes (an optical fiber or a metallic

tip) and the nanoresonator can lead to spectral shifts of the modes and potential distortions when probing the anapoles. These distortions are not observed when employing EELS, and thus, we can access to “clean anapoles” which are hardly accessible with near-field optical techniques.

- Spatial resolution: By using EELS, we are able to acquire energy filter images of the anapoles with spatial resolutions below 1 nm. In near-field microscopy, however, the spatial resolution is usually on the order of tens of nanometers, and it is limited by the diameter of the optical fiber in aperture-based SNOM or by the tip-apex radius in apertureless SNOM.
- Broadband acquisition: Fast electron beams are white sources of electromagnetic fields that allow us to obtain spectral information about a sample over a wide spectral range, from the near infrared up to the ultraviolet.

In the revised manuscript, we explicitly emphasize the advantages of using EELS over far- and near-field optical spectroscopy and microscopy by adding the following lines to the introduction:

... **Complementary to far-field characterization, understanding how anapoles are excited by localized probes can be of paramount importance in order to control and realize the full potential of these non-radiating states. In this context, Scanning Near-Field Optical Microscopy (SNOM) was applied to study the near-field distribution of Si disks at relevant wavelengths [5, 9]. In this first experiment, the field around the disk was mapped at the anapole wavelength, revealing a maximum in the amplitude of the near field in the middle of the disk [5]. In the second experiment, it was found that the normal component of the electric field produced by the disks is reduced at the anapole wavelength [9]. SNOM offers the advantage of spatially mapping both the amplitude and phase of near-field patterns, however, the spatial resolution is limited by the dimensions of the near-field probes (an optical fiber or a metallic tip), which are typically of the order of tens of nanometers. More importantly, the near-field probe itself can couple to the sample [17], leading to spectral shifts of the resonance modes and potential distortions when probing the anapoles. In contrast to SNOM, Electron Energy Loss Spectroscopy (EELS) in Scanning Transmission Electron Microscopy (STEM) is a non-disturbing technique which accesses to the clean electromagnetic fields and modes of a sample. At the same time, EELS allows access not only to electric dipole modes but also quadrupoles and higher-order modes that do not, or weakly, couple to far-field radiation.**

Comment No. 2

Why the authors choose WS₂ nanodisks to study optical anapole, not silicon?

Response: Tungsten disulfide has better mechanical and optical properties than silicon, it is simpler to fabricate the nanodisks from multilayered WS₂, and we had previous experience with this material. In addition, the presence of the A-exciton in WS₂ offers the advantage of exploring anapole-exciton hybridization within the same nanoobject, a situation already emphasized in the original manuscript.

To emphasize why we use WS₂, we added the following paragraph to the introduction of the revised manuscript:

Optical anapoles are typically found in high-index dielectric nanostructures. To experimentally access the spectral and spatial information on the anapoles, we fabricate nanodisks from exfoliated multilayer TMDC WS₂ with a high refractive index in the visible and infrared ranges [32-34], even larger than that of standard Si and other semiconductors, usually employed to fabricate anapole resonators.

In Fig.1b, the authors calculated EEL probability mapping and assign the dip as anapoles. I may suggest the authors also include the scattering cross section mapping to double confirm the existence of anapoles because it is the well-established way to define the anapoles.

Response: Following this suggestion, we include the scattering cross section mapping in the main text. Along the lines of the first comment by this reviewer, this new figure emphasizes the differences (and advantages) between using electron beams and far-field illumination.

In the revised manuscript, we have included a new figure (Fig. 2) showing the scattering cross sections of the high-index dielectric disks with $\varepsilon = 18$. To confirm the appearance of anapoles, we have also included in the new Fig. 2 the multipole decomposition of the induced current density inside the disk, together with the near-field patterns at those energies where the anapoles appear.

In the following we show the new Fig. 2:

FIG. 2. Optical anapoles excited by plane wave illumination. (a) A high-index dielectric nanodisk (blue cylinder) is excited by a plane wave (E_{inc} , red arrow) that propagates perpendicular to the top surface of the disk (normal incidence) and is polarized along the x -axis. (b) Simulated scattering cross section $\sigma_{\text{scat}}(\omega)$ of the nanodisk calculated as a function of the disk radius R and the photon energy $\hbar\omega$. Gray dashed lines are guides to the eye and indicate the position of the first three dips in the scattering cross section spectra. (c) $\sigma_{\text{scat}}(\omega)$ obtained for the nanodisk with $R = 250$ nm. The red dots mark the spectral dip positions. (d) Partial scattering cross section of the P_{sph} , P_{car} and T_{car} dipole moments induced in the nanodisk with $R = 250$ nm. (e) Partial scattering cross section of the \hat{Q}_{sph} , $\hat{Q}_{\text{car}}^{(e)}$ and $\hat{Q}_{\text{car}}^{(T)}$ quadrupole moments induced in the nanodisk with $R = 250$ nm. (f) Amplitude of the total electric field $|E(\omega)|$ inside the disk at the plane $z = 0$, for energies: (A_{11}^E) 1.255 eV, (A_{12}^E) 1.87 eV and (A_{13}^E) 2.46 eV. The scale bar is 100 nm. The field plots are normalized to the amplitude of the incident plane wave $|E_{\text{inc}}|$. In each case, the maximum value of $|E(\omega)|/|E_{\text{inc}}|$ is: 3.0, 5.4 and 4.7.

In the revised manuscript, we discuss the scattering cross section spectra in the following paragraphs:

To show the advantage of EELS over far-field optical spectroscopy in probing anapoles, we compare

the EEL spectra of the high-index dielectric disks with the far-field optical scattering spectra of the same disks. To that end, we calculate the scattering cross section $\sigma_{\text{scat}}(\omega)$ of the disks shown in Fig. 1(b) illuminated with a linearly-polarized plane wave propagating along the z -axis (illustrated in Fig. 2(a)). The result is shown in Fig. 2(b) and (c) (see Methods and Supplementary Note 1 for the details of the numerical simulations). Similar to the EEL spectra $\Gamma(\omega)$, we observe dips in $\sigma_{\text{scat}}(\omega)$ that shift to higher energies as the nanodisk radius R decreases (see gray dashed lines in Fig. 2(b)). By performing a multipole decomposition of the current density $\mathbf{J}_{\text{ind}}(\omega)$ induced in the disk with $R = 250$ nm (see Figs. 2(d) and (e)), we can associate these dips (analogue to the discussion of Fig. 1) with the excitation of the first (A_{11}^E), second (A_{12}^E) and third (A_{13}^E) electric dipole anapole states, whose field distributions inside the disk are shown in Fig. 2(f). On the other hand, quadrupolar contributions are negligible, compared to dipolar contributions (Figs. 2(e) and (d)), highlighting the advantage of EELS for probing higher-order anapole states.

Comment No. 4

For Fig.3a, it is very hard to differentiate the strong coupling because avoided-crossing is not obvious. I may suggest authors presenting the EEL spectrum at different radius in the supplementary materials, which may help readers better observe strong coupling.

Response: We thank the reviewer for this suggestion. We have included a new Supplementary Note (Supplementary Note 8), where we plot the experimental EEL spectra for the different radius of the disk:

Supplementary Figure 9. Experimental EEL spectra of the nine disks displayed in Fig. 3 of the main text. The spectra are the same as the ones shown in Fig. 4(a) of the main text but offset for better visualization. The green dashed line indicates the A-exciton frequency (see Supplementary Note 10).

We added the following sentences to the caption of Fig. 4:

For better visualization, we show in Supplementary Note 8 the experimental EEL spectra for each disk radius.

Comment No. 5

In the experimental section, the authors define “impact parameter ” without any explanation, please add the description on it.

Response: We now include the definition of the impact parameter.

Paragraph 24: “To that end, we collect the EEL signal as a function of the position of the fast electron with respect to the nanodisk center (impact parameter b)”

Comment No. 6

Finally, some important references related to anapoles and TMDC bulks are missing.

- (a) Application of anapoles in THG, L. Xu, et al, “Boosting third-harmonic generation by a mirror-enhanced anapole resonator”, *Light: Science & Applications* 7,44(2018).
- (b) High order anapoles, E. A. Gurvitz, et al “The High-Order Toroidal Moments and Anapole States in All-Dielectric Photonics”, *Laser Photonics Reviews* 13(5),1800266(2019).
- (c) V. A. Zenin,et al, “Direct Amplitude-Phase Near-Field Observation of Higher-Order Anapole States”, *ACS Nano* 17,7152(2017).
- (d) TMDC Bulk, S. Busschaert, et al “Transition Metal Dichalcogenide Resonators for Second Harmonic Signal Enhancement”, *ACS Photonics* 7, 2482(2020).
- (e) L. Huang et al, “Enhanced light–matter interaction in two-dimensional transition metal dichalcogenides”, *Rep.Prog. Phys* 85, 046401(2022).

Response: Reference (c) has already been cited in the original manuscript. We have included in the right context the rest of the references suggested by the reviewer.

RESPONSE TO REVIEWER 2

General Assessment

The results presented in this work are intriguing and will be useful to the community. However, the paper is not well written and the results are not always clearly presented. In addition, certain key questions related to the validity of the results are not addressed.

Response: We thank the reviewer for her/his appreciation of our work. We have tried hard to improve the quality of both the manuscript and the Supplementary Information.

Comment No. 1

Figure 1 is misleading. Panel (a) shows the disk supported by a substrate, but the substrate is not taken into account in the calculations of panels (b-e).

Response: We agree with the reviewer that panel (a) in Fig. 1 is misleading. Accordingly, we removed the gray box in Fig. 1(a).

Comment No. 2

Also in Fig. 1: Panel (f) illustrates the formation of anapole modes arising from the interference of electric dipole and toroidal dipole and from the interference of electric quadrupole and toroidal quadrupole. The resulting anapole modes are termed “First ED anapole” and “First QD anapole”, which implies that that “Second ED anapole” is formed differently. The authors should clarify this point.

Response: The reviewer is correct. The First ED and the First EQ anapoles are formed differently from the Second ED anapole. The latter is the result of the far-field interference of the radiation produced by the electric dipole moment \mathbf{P}_{car} , the toroidal dipole moment \mathbf{T}_{car} , and higher-order toroidal dipole moments. The theoretical description of higher-order toroidal moments and their influence on the anapole formation has been previously reported by “Gurvitz, *et al. Laser Photonics Rev.* 13, 1800266 (2019)”. The comment of the reviewer made us aware that in the original manuscript we did not explicitly mention the differences between higher-order anapoles, and labeling the anapole dips as “A”, “B”, “C”, and “D” was not precise. Thus, we have changed them to the more precise labels A_{ij}^E , which denote the j -th dip of the scattered power from the electric spherical 2^i -pole.

We have added the following comment to the revised manuscript:

The subsequent dips in the EEL spectrum (marked A_{12}^E and A_{22}^E in Fig. 1(c)) are associated with the excitation of higher-order anapole states such as the second electric dipole anapole (marked A_{12}^E) and the second electric quadrupole anapole (marked A_{22}^E) states. These dips originate from the destructive interference of the radiation produced by Cartesian electric and higher-order toroidal multipoles excited in the nanodisk [40]. The contribution of these higher-order multipoles is larger at higher energies (shorter wavelengths), and thus the conventional Cartesian multipole decomposition in the long-wavelength approximation fails to describe the spectral positions of A_{12}^E and A_{22}^E (compare the position of the red dots and the grey dashed lines above 1.75 eV in Figs. 1(d) and (e)). The spherical multipole decomposition, on the other hand, very accurately reproduces the dips (anapole states) appearing in the EEL spectra. This allows us to label the j -th dip of the scattered power from the electric spherical 2^i -pole as A_{ij}^E .

Comment No. 3

It is difficult to compare the contributions of different multipoles as the units are different. The scattered power from each multipole should be used instead. See for instance doi: 10.1103/PhysRevE.65.046609.

Response: We have implemented the suggestion of the reviewer. In the revised manuscript and Supplementary Information, instead of plotting the amplitude of the electric dipole and quadrupole moments, we now plot the scattered power from each multipole which were calculated using the following expressions:

$$P_{\text{scat}}^{\mathbf{P}^{\text{sph}}}(\omega) = \frac{k_0^4 c}{12\pi\epsilon_0} |\mathbf{P}_{\text{sph}}(\omega)|^2, \quad P_{\text{scat}}^{\mathbf{P}^{\text{car}}}(\omega) = \frac{k_0^4 c}{12\pi\epsilon_0} |\mathbf{P}_{\text{car}}(\omega)|^2, \quad P_{\text{scat}}^{\mathbf{T}^{\text{car}}}(\omega) = \frac{k_0^4 c}{12\pi\epsilon_0} |ik_0 \mathbf{T}_{\text{car}}(\omega)|^2, \\ P_{\text{scat}}^{\hat{\mathbf{Q}}^{\text{sph}}}(\omega) = \frac{k_0^6 c}{1440\pi\epsilon_0} \left| \hat{\mathbf{Q}}_{\text{sph}}(\omega) \right|^2, \quad P_{\text{scat}}^{\hat{\mathbf{Q}}^{\text{car}}^{(e)}}(\omega) = \frac{k_0^6 c}{1440\pi\epsilon_0} \left| \hat{\mathbf{Q}}_{\text{car}}^{(e)}(\omega) \right|^2 \quad \text{and} \quad P_{\text{scat}}^{\hat{\mathbf{Q}}^{\text{car}}^{(T)}}(\omega) = \frac{k_0^6 c}{1440\pi\epsilon_0} \left| i3k_0 \hat{\mathbf{Q}}_{\text{car}}^{(T)}(\omega) \right|^2.$$

These were obtained from Refs. “Radescu, *et al. Phys. Rev. E* 65, 046609 (2002)” and “Alaee, *et al. Opt. Commun.* 407, 17-21 (2018)”.

In the revised manuscript, we have changed and homogenized the units to those of the scattered power (Watts). We explain the details of the calculations in the added Supplementary Note 4.3.

Comment No. 4

The authors take into account only four multipoles: electric dipole, toroidal dipole, electric quadrupole, and toroidal quadrupole. They should calculate all multipoles up to toroidal quadrupole order and provide evidence that these multipoles adequately represent the excitations of the disk.

Response: We have performed numerical calculations of the scattered power from the Cartesian multipoles up to the toroidal quadrupole moment, using the following standard expression:

$$P_{\text{scat}}^{\text{car}}(\omega) = \frac{k_0^4 c}{12\pi\epsilon_0} |\mathbf{P}_{\text{car}}(\omega) + ik_0 \mathbf{T}_{\text{car}}(\omega)|^2 + \frac{k_0^6 c}{1440\pi\epsilon_0} \left| \hat{\mathbf{Q}}_{\text{car}}^{(e)}(\omega) + i3k_0 \hat{\mathbf{Q}}_{\text{car}}^{(T)}(\omega) \right|^2. \quad (\text{R.1})$$

For comparison, we also evaluated the scattered power resulting from the spherical dipole and quadrupole moments using the next expression:

$$P_{\text{scat}}^{\text{sph}}(\omega) = \frac{k_0^4 c}{12\pi\epsilon_0} |\mathbf{P}_{\text{sph}}(\omega)|^2 + \frac{k_0^6 c}{1440\pi\epsilon_0} \left| \hat{\mathbf{Q}}_{\text{sph}}(\omega) \right|^2. \quad (\text{R.2})$$

We present both results in the figure for review, Fig. R.1, where we additionally plot the EEL probability. The numerical calculations were performed for a nanodisk with $R = 250$ nm, $d = 55$ nm and $\epsilon = 18$ excited by an electron beam traveling in an aloof trajectory along the z -axis at a distance $b = 1.1R$ with respect to the nanodisk center (same as the ones shown in Fig. 1(c)-(e) of the main text). Each spectrum is normalized to the first peak maximum.

From Fig. R.1, we can observe that the EEL spectrum (blue line) is adequately reproduced by the scattered power from the Cartesian (black line obtained with Eq. R.1) and spherical (red line obtained with Eq. R.2) multipole moments within the energy range of 0.5 eV to 1.5 eV. Above this energy, the EEL spectrum exhibits some features that are not captured using Eqs. R.1 and R.2, such as peak amplitudes, linewidths, and the number of peaks. However, it is possible to reproduce these spectral features by considering the scattered power from higher-order multipoles, including the octupole, 2^4 -pole, and subsequent orders, into Eqs. R.1 and R.2.

Despite these differences, we stress that the spectra obtained with the scattered power from the multipoles up to the quadrupole moment fully recovers some of the dips in the EEL spectrum (compare the three spectra at the energies marked by the grey dashed lines). This supports our claim and provides evidence that dipole and quadrupole anapole states are excited by the fast electron beam.

Figure R.1. Comparison between the electron energy losses and the scattered powers from the multipoles. EEL probability (blue line) of the high-index dielectric disk with $\epsilon = 18$, $R = 250$ nm and $d = 55$ nm, as obtained from numerical simulation for an aloof electron beam traveling with velocity $v = 0.7c$ and impact parameter $b = 1.1R$. The black and red lines are the scattered powers from the Cartesian (Eq. R.1) and spherical (Eq. R.2) multipole moments, respectively. Each spectrum is normalized to the first peak maximum. The gray dashed lines mark the positions of the anapole states.

We added the following paragraph in the revised manuscript:

To compare the multipole decomposition with the EEL spectrum of the 250 nm disk, we calculate the scattered power from the dipole and quadrupole moments of the induced current density $\mathbf{J}_{\text{ind}}(\mathbf{r})$ (see Supplementary Note 4). The scattered power from these multipole orders adequately captures the spectral features of the EEL spectrum across the energy range from 0.5 eV to 1.5 eV. Above 1.5 eV the EEL spectrum reveals additional spectral features that can be reproduced by also considering the scattered power from higher-order multipoles.

Comment No. 5

The reference list does not adequately represent the field. The term “anapole” was first introduced (in the static regime) by Zel’dovich, in *J. Exp. Theor. Phys.* 33, 1184–1186 (1957). The first experimental observation of a dynamic anapole was by Fedotov et al. in <https://doi.org/10.1038/srep02967>.

Response: We thank the reviewer for pointing this out. We have included in the revised manuscript these seminal references in the right context.

Comment No. 6

How important is the anisotropy of the permittivity tensor in the excitation of anapole modes?

Response: In our specific case, the anisotropy of WS_2 has no influence on the anapole excitation in the nanodisk. To demonstrate this, we first calculate the EEL probability for a nanodisk with anisotropic permittivity of $\hat{\epsilon} = \text{diag}(18, 18, 7)$ and compare its spectrum with that obtained for a nanodisk with isotropic permittivity of $\hat{\epsilon} = \text{diag}(18, 18, 18)$. We find that the use of the anisotropic permittivity has no effect on the spectral dips in EEL spectra. Additionally, we compare the scattered powers from Cartesian electric and toroidal dipoles induced in both disks. We verify that the interference between Cartesian electric and toroidal dipoles occur at the same spectral positions for both isotropic and anisotropic disks. We check that these spectral positions correspond precisely to the dips observed in the EEL spectra. This supports our claim that anisotropy does not play a role in influencing the

excitation of anapole states in the disk.

In the revised manuscript, we added the following comment:

... The differences between the solid and the thin blue spectra in Fig. 4(e) are a direct consequence of the appearance of the A-exciton resonance at 1.96 eV (identified in the EEL spectrum of a WS₂ flake shown by the green lines in Fig. 4(e)). We note that multilayer WS₂ is a natural anisotropic material, however, the anisotropy in this case does not influence the anapoles excitation, and the spectral response of the disk with isotropic permittivity is nearly identical to that of an anisotropic disk (see Supplementary Note 9).

In the revised Supplementary Information, we report the results described above in a new Supplementary Note 9, where we show the following figure with the corresponding explanatory text:

Supplementary Figure 10. Anisotropic effects. (a) Comparison of simulated EEL spectra for the anisotropic (green dashed line) and isotropic (black line) permittivity. The numerical calculations were performed for a nanodisk with radius $R = 250$ nm and thickness $d = 55$ nm excited by an electron beam traveling with velocity $v = 0.7c$ in a aloof trajectory along the z -axis at a distance $b = 1.1R$ with respect to the nanodisk center. The red dots and the gray dashed lines mark the anapole dips A_{11}^E and A_{12}^E . (b) Scattered power of the dipole moments induced in the anisotropic nanodisk. (c) Scattered power of the dipole moments induced in the isotropic nanodisk.

The authors should discuss the dependence of the excited modes on the electron velocity.

Response: We agree with the reviewer that this is an important point to discuss. For this reason, we numerically calculate the electron energy loss probability, $\Gamma(\omega)$, for different electron velocities $v = 0.8c, 0.7c, 0.6c, 0.5c$ and $0.4c$. In the calculations, we consider a high-index dielectric ($\epsilon = 18$) nanodisk with radius $R = 250$ nm and thickness $d = 55$ nm excited by an electron beam traveling close to the nanodisk in an aloof trajectory at the fixed impact parameter $b = 1.1R$.

From the simulations, one clearly observes that the position of the peaks and dips appearing in the EEL spectra do not change for different electron velocities. These features indicate that, for a fixed impact parameter, neither disk modes nor anapole excitation are significantly influenced by variations in the velocity of the electron.

In order to clarify the dependence of the excited modes and anapoles on the electron velocity, we added a new Supplementary Note (Supplementary Note 2), where we show the following figure with the corresponding explanatory text:

Supplementary Figure 3. Theoretical EEL spectrum for different electron velocities v . Comparison of simulated EEL spectra for $v = 0.8c, 0.7c, 0.6c, 0.5c$ and $0.4c$, as indicated in the inset. The numerical calculations were performed for a nanodisk with $R = 250$ nm, $d = 55$ nm and $\epsilon = 18$ excited by an electron beam traveling in an aloof trajectory along the z -axis at a distance $b = 1.1R$ with respect to the nanodisk center. The gray dashed lines mark the four anapole dips $A_{11}^E, A_{21}^E, A_{12}^E$ and A_{22}^E . Yellow spectrum is the same as the one shown in Fig. 1(c) of the main text.

In the revised manuscript, we added the following sentence:

Figure 1(b) shows the calculated $\Gamma(\omega)$ spectra for different nanodisk radius R in the energy range of 0.5 eV to 2.5 eV. The spectra feature a series of peaks (white dotted lines) and dips (gray dashed lines) that monotonously shift to higher energies as the nanodisk radius R reduces from 300 nm to 100 nm. The positions of these peaks and dips do not vary with the electron beam velocity (see Supplementary Note 2).

The samples studied experimentally include a thin substrate. Can the authors discuss the role of the substrate in the anapole excitations? For instance, are the electromagnetic fields of the resonant modes mainly confined in the disk or is there substantial excitation of the substrate too?

Response: The SiN substrate redshifts the resonant modes of the nanodisk, and thus, the anapole dips. Which are caused by the interference of the resonant modes.

The electromagnetic field associated with the resonant modes is predominantly confined within the disk. Notably, for higher-order modes and anapoles, the field exhibits increased localization inside the resonator, leading to a reduced influence of the SiN substrate. A similar observation has already been documented in Ref. “Zenin, *et al. Nano Lett.* 17, 11 (2017)”.

To clarify the effects of the substrate on the anapoles excitation, we added a new Supplementary Note 11, where we show the following figure with the corresponding explanatory text:

Supplementary Figure 13. Influence of the substrate. (a) Comparison of EEL spectra with (blue line) and without (blue shaded curve) a thin isotropic substrate of 50 nm thickness and constant permittivity $\epsilon_{\text{SiN}} = 4.1853$. The numerical calculations were performed for a nanodisk with radius $R = 250$ nm and thickness $d = 55$ nm excited by an electron beam traveling with velocity $v = 0.7c$ in an aloof trajectory along the z -axis at a distance $b = 1.1R$ with respect to the nanodisk center. The red dots and the red dashed lines mark the anapole dips A_{11}^E , A_{21}^E , A_{12}^E and A_{22}^E . The black dashes lines mark the peak positions at the energies 1.332 eV, 1.616 eV and 1.86 eV. (b) Scattered power of the dipole moments in the disk on top of a substrate. (c) Scattered power of the quadrupole moments in the disk on top of a substrate. (d)-(i) Amplitude of the total electric field $|E(\omega)|$ at the plane $x = 0$ (depicted in the schematics above the field plots) for the energies marked by the dashed lines in panel (a). The field plots are normalized to the maximum value $|E_{\text{max}}|$ in each case and the scale bar is 100 nm.

In the revised manuscript, we added the following sentences to the Methods section (Numerical simulations subsection):

For simplicity, numerical calculations shown in Figs. 1 and 2 were performed without considering any

substrate, whereas in Figs. 4 and 5 the nanodisk is on top of a 50 nm thick substrate layer characterized by the permittivity of SiN $\epsilon_{\text{SiN}} = 4.1853$. We note that the SiN substrate does not alter the excitation of the anapole states. It slightly redshifts the resonant modes of the nanodisk, and thus, the anapole dips simply appear at lower energies (see Supplementary Note 11).

Comment No. 9

In the Supplementary material, in Eq. (1) the limits of the integration refer to parameter L_{PML} , which is not defined.

Response: In the original manuscript, the parameter L_{PML} was defined in the Methods section (Numerical Simulations subsection), and for this reason, we did not include its definition in the Supplementary Information (SI). However, to make the SI self-consistent, we added the definition of L_{PML} in the first paragraph of revised Supplementary Information:

Paragraph 1: “...where e is the elementary charge, L_{PML} is the length of the simulation box (equal to $12 \times R$, R the disk radius), $\text{Re}[x]$ represents the real part of the complex number x , $E_z^{\text{ind}}(\mathbf{r})$ is the z -component of the induced electromagnetic field in the nanodisk and $E_z^{\text{ind}}(\mathbf{r})$ is evaluated along the trajectory of the electron beam $\mathbf{r}_e(t) = (x_e = b, y_e = 0, z = vt)$.”

RESPONSE TO REVIEWER 3

General Assessment

The authors have used Scanning Transmission Electron Microscopy (STEM) to probe optical anapoles in WS₂ nanodisks. Most characterization of anapole modes were conducted by using optical measurements so the use of electron energy loss spectroscopy seems to be novel. It may be useful for a small number of people but I don't believe it will reach a broad audience. The methodology is sound for both theoretical and experimental analyses. Also, the theoretical and experimental results have a good match.

Response: We thank the reviewer for her/his appreciation of our work. We believe that our findings will be of critical importance and of great interest to the multidisciplinary research community working on high-index dielectric structures and dark-scattering states, where appropriate spectroscopic techniques are required to probe and map the rich optical response of the dielectric structures. We emphasize that, as mentioned in the final sentence of Ref. "Savinov, *et al. Commun. Phys.* 2, 69 (2019)", this community has envisioned the excitation and probing of optical anapoles using fast electron beams.

We think our work will be also relevant to the audience interested in TMDC materials and their potential as optical platforms. The field of nanophotonics based on TMDCs is relatively new, but it is rapidly expanding and continuously discovering novel structures promising to enrich the existing library for nanophotonics and polaritonics applications. Notable examples of this include the observation of highly efficient second harmonic generation in WS₂ disks ("Busschaert, *et al. ACS Photonics* 7, 9 (2020)"), the identification of Mie resonances in MoS₂ disk arrays ("Shen, *et al. Nat. Commun.* 13, 5597 (2022)"), and the recent finding of intrinsic strong coupling between bound states in the continuum and excitons in metasurfaces composed of WS₂ ("Weber, *et al. Nat. Mater.* (2023)").

We anticipate that our results will impact a broad audience investigating, mapping, and characterizing various optical excitations taking place in high-index resonators and TMDC materials. Thus, we strongly believe our results target a broad audience.

Reviewers' Comments:

Reviewer #1:

Remarks to the Author:

The authors has addressed all of my concerns. I am happy to recommend it for publications in Nature Communications.

Reviewer #2:

Remarks to the Author:

The authors have revised their manuscript and have answered most of issues raised during the first round of review. Before recommending the paper for publication, I would like to see a revised version of the manuscript where the authors clarify the following:

1. It is not clear how spectral features are attributed to higher order anapoles. For instance, can they distinguish between A12 and A13 anapoles?
2. The relation between higher order anapoles and mean square radii needs to be discussed, see <https://doi.org/10.1103/PhysRevA.98.023858>.
3. In the multipole calculations of Suppl. Fig. 13: Do the authors take into account the fields inside the substrate while calculating the different multipole contributions to scattering? If not, how can they be certain that the features in the EELS spectra are not due to scattering from the substrate?

Probing optical anapoles with fast electron beams

Response letter to the second reviewer's report

Carlos Maciel-Escudero, Andrew B. Yankovich, Battulga Munkhbat, Denis G. Baranov, Rainer Hillenbrand, Eva Olsson, Javier Aizpurua, and Timur O. Shegai
(Dated: October 24, 2023)

We thank again the reviewers for their comments and efforts that helped to improve the clarity of our manuscript. In the following, we reproduce in full the comments by the reviewers within the boxes. We provide our answer after each box, and the amendments made to the manuscript are highlighted in blue.

RESPONSE TO REVIEWER 1

General Assessment

The authors has addressed all of my concerns. I am happy to recommend it for publications in Nature Communications.

Response: We thank the reviewer for acknowledging our work.

RESPONSE TO REVIEWER 2

Comment No. 1

The authors have revised their manuscript and have answered most of issues raised during the first round of review. Before recommending the paper for publication, I would like to see a revised version of the manuscript where the authors clarify the following.

It is not clear how spectral features are attributed to higher order anapoles. For instance, can they distinguish between A12 and A13 anapoles?

Response: We thank the reviewer for her/his appreciation of our hard work in addressing comments and suggestions raised in the first round of the review. We are also happy to address her/his additional comments.

As we show in Fig. 2(f) of the original manuscript, anapoles A_{11}^E and A_{12}^E exhibit distinct spatial field distributions, allowing them to be distinguished. In order to emphasize this distinction, we have incorporated the following phrase into the revised manuscript:

By performing a multipole decomposition of the current density $J_{\text{ind}}(\omega)$ induced in the disk of $R = 250$ nm (see Figs. 2(d) and (e)), we can associate these dips (analogue to the discussion of Fig. 1) with the excitation of the first (A_{11}^E), second (A_{12}^E) and third (A_{13}^E) electric dipole anapole states. The differences between these anapoles can be observed by plotting their spatial field distributions, as shown in Fig. 2(f). We observe that the A_{11}^E anapole exhibits the characteristic field distribution of the first electric dipole anapole state. The A_{12}^E and A_{13}^E anapoles, however, exhibit additional nodes at the edges of the disk, originating from resonant modes with higher radial order, i.e., higher number of nodes along the radial direction.

Comment No. 2

The relation between higher order anapoles and mean square radii needs to be discussed, see <https://doi.org/10.1103/PhysRevA.98.023858>.

Response: As requested by the reviewer, we have added in the revised manuscript the following discussion, together with the suggested reference:

These dips originate from the destructive interference of the radiation produced by Cartesian electric and higher-order toroidal multipoles excited in the nanodisk [40]. These higher-order multipoles are connected with additional higher-order terms, and can be also referred to as the mean-square radii of the respective multipoles [41].

Comment No. 3

In the multipole calculations of Suppl. Fig. 13: Do the authors take into account the fields inside the substrate while calculating the different multipole contributions to scattering? If not, how can they be certain that the features in the EELS spectra are not due to scattering from the substrate?

Response: In the calculations of the multipole decomposition shown in Supplementary Fig. S13, we have not considered the fields inside the SiN substrate. However, to confirm that the spectral features in the EEL spectra are not due to scattering from the substrate, we numerically calculate its reflectivity.

We present the result in the figure for review, Fig. R.1, where we plot the reflectivity of a 50 nm thick layer characterized by the permittivity of SiN $\epsilon_{\text{SiN}} = 4.1853$. For comparison, we also plot the reflectivity of a 70 nm thick layer made of WS₂, mimicking the WS₂ disks. The numerical calculations are obtained with the transfer-matrix formalism discussed in Supplementary Note 6.

From Fig. R.1, we observe that within the energy range of 1.0 eV to 1.6 eV, the reflectivity of WS₂ (red curve) is at least 2.5 times higher than the reflectivity of SiN (blue curve). This supports our claim that the dips and peaks that appear in the EEL spectra, within the energy range of 1.0 eV to 1.6 eV, are predominantly a result of the optical excitations that occur in the WS₂ disk.

Figure R.1. Comparison between the reflectivity of SiN and WS₂. Reflectivity of a SiN layer of 50 nm thickness (blue line) and a WS₂ layer of 70 nm thickness (red line). The numerical calculations are obtained with the transfer-matrix formalism and dielectric functions reported in Supplementary Note 6.

In addition, we want to stress that the excitation of optical anapoles by fast electron beams is demonstrated in Fig. 1 of the original manuscript, where we show the EEL spectra and the multipole decomposition without considering any substrate in the calculations. This further corroborates that the important spectral features in the EEL spectra are not caused by the SiN substrate.

In order to clarify the role of the substrate in the spectral features of the EEL spectra, we added the following paragraph in the revised manuscript:

In addition, we confirm that the reflectivity of WS₂, within the energy range of 1.0 eV to 1.6 eV, is at least 2.5 times higher than the reflectivity of SiN. This allows us to discard substrate effects and also corroborates that the dips and peaks in the EEL spectra are predominantly due to optical excitations of the WS₂ disk.

Reviewers' Comments:

Reviewer #2:

Remarks to the Author:

The authors have addressed the issues raised and I am happy to recommend the paper for publication

20.11.2023

Reviewer #2 (Remarks to the Author): *The authors have addressed the issues raised and I am happy to recommend the paper for publication.*

We thank all the reviewers for the constructive criticism of our work.

Sincerely,
Prof. Timur Shegai
Prof. Javier Aizpurua
Prof. Eva Olsson